# Position: Safe Models Do Not Guarantee Safe Societies

**David Guzman Piedrahita** [1 2 3]  **Changling Li** [1 2 4]  **Dave Banerjee** [5]  **Terry Jingchen Zhang** [1 2 3]  **Kevin Blin** [1 3]
**Samuel Simko** [1 2 3]  **Punya Syon Pandey** [1 3]  **Irene Strauss** [2]  **Rada Mihalcea** [6]  **Bernhard Schölkopf** [4 7]
**Zhijing Jin** [1 4 3]

## Abstract

Sociopolitical AI risks are threats to collective self-determination: a society's capacity to articulate its interests and realize them through institutions. We argue that sociopolitical AI risks emerge when general-purpose AI systems are integrated into society in ways that disproportionately amplify the scale, speed, and opacity of institutional operations, thereby degrading their capacity to function. Unlike model-level harms (toxicity, bias, discrimination), sociopolitical risks arise from widespread deployment rather than individual outputs. And unlike existential risks involving loss of control or complete labor automation, they manifest with current AI capabilities where AI augments rather than replaces human activity. In this position paper, we analyze how AI alters the conditions of governance via flooding government agencies with paralyzing volumes of input, concentrating control of infrastructure that threatens sovereignty, and flattening public debate into artificial agreement while reinforcing biases. We conclude with recommendations for evaluation methodology, institutional design, and procurement policy that treat sociopolitical resilience as a first-class objective alongside model-level safety.

## 1. Introduction

In January 2024, AI-generated robocalls impersonating then-President Biden reached an estimated 25,000 New Hampshire voters days before the presidential primary, urging them to "save their vote" for November (Atherton, 2024; Swenson & Weissert, 2024). The spoofed caller ID of a local official made the call look legitimate, and the synthetic voice was nearly indistinguishable from the real one. Deceptive tactics in politics are nothing new, but AI has sharply reduced the cost of deploying them, while the institutional mechanisms for attribution and correction have not scaled at the same rate (Bontridder & Poullet, 2021). This asymmetry between the falling cost of AI-generated influence and the persistent cost of institutional response extends well beyond election interference: it reshapes how citizens participate, how agencies process public input, and how societies maintain shared understanding. **In this position paper, we argue that these dynamics constitute a distinct class of AI risks, *sociopolitical risks*, that emerge from the integration of general-purpose AI into social and political systems at scale, and that this class cannot be fully resolved through model-level alignment.** Some persist regardless of alignment quality, as a voice-cloning robocall deceives whether or not the underlying model is well-aligned, others arise directly from how current alignment methods work, such as training for user approval, and only some are tractable through improving model behavior alone.

AI alignment methods focus on model-level properties such as bias and toxicity (Weidinger et al., 2021), while broader AI safety discourse emphasizes existential risks involving sudden loss of control and catastrophic misuse (Carlsmith, 2022; Hendrycks et al., 2023). Recent work on gradual disempowerment characterizes the macro arc through which AI may erode human collective agency (Kulveit et al., 2025). None of these frameworks is designed to capture how AI integration into courts, regulatory agencies, electoral systems, and public-comment processes degrades the institutional conditions on which democratic governance depends. While prior work on digital platforms has documented how algorithmic systems can distort public discourse (Zuboff, 2019; Benkler et al., 2018; Pasquale, 2015), general-purpose AI introduces qualitatively new dynamics: a single model now mediates across domains that previously required separate systems, embedding value commitments at scale while making content far cheaper to produce than to verify.

We propose **sociopolitical AI risks** as a framework to cap-

---

[1]EuroSafeAI [2]ETH Zürich [3]Jinesis Lab, University of Toronto & Vector Institute [4]Max Planck Institute for Intelligent Systems, Tübingen, Germany [5]Institute for AI Policy and Strategy [6]University of Michigan [7]ELLIS Institute Tübingen. Correspondence to: David Guzman Piedrahita <david.guzmanpiedrahita@uzh.ch>, Changling Li <changlingli.mpi@gmail.com>, Zhijing Jin <zjin@cs.toronto.edu>.

*Proceedings of the 43rd International Conference on Machine Learning*, Seoul, South Korea. PMLR 306, 2026. Copyright 2026 by the author(s).

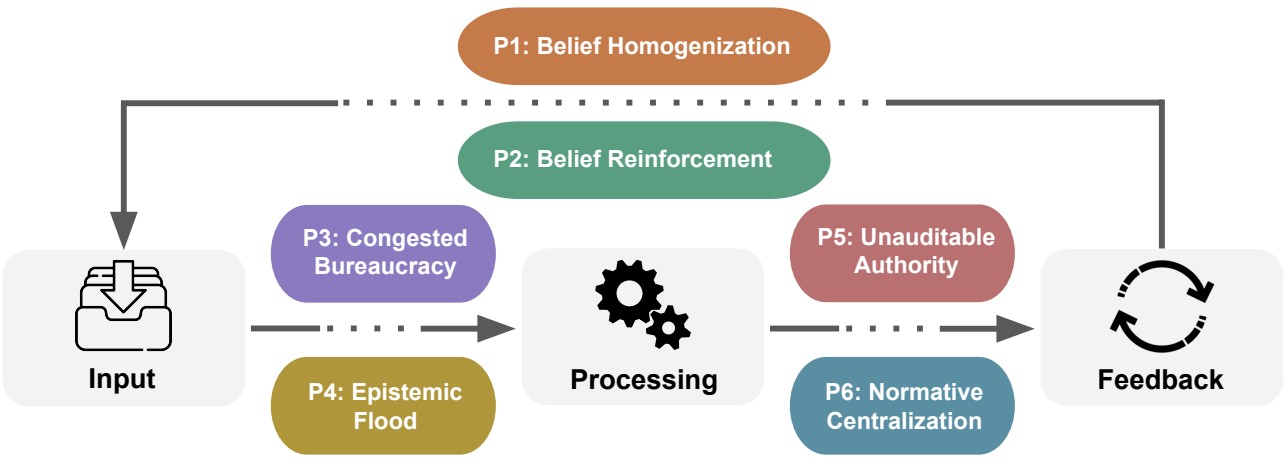

*Figure 1.* Governance as an information-processing loop (Input → Processing → Feedback → Input). Threat models are positioned at the boundary where they primarily weaken responsiveness, contestability, or belief updating.

ture this gap: threats to a society's capacity to articulate collective interests and realize them through accountable democratic institutions. Unlike alignment failures addressable through model-level fixes, sociopolitical risks emerge from aggregate deployment effects. A single AI-generated output may be harmless; in aggregate, many such outputs can flood public-comment systems, mislead officials, or make government decisions harder to appeal (Figure 1). These failures manifest with current AI capabilities, where AI augments rather than replaces human activity, and persist even if models are well-aligned (Christiano, 2018).

AI researchers should care about these dynamics because they compromise the institutions on which safety governance itself depends. If public consensus and regulatory capacity are eroded by the very tools those institutions are meant to oversee, society loses its ability to coordinate responses to catastrophic risks (Acemoglu, 2024). At the same time, model-level safety depends on functioning institutions to define what "safe" means and to hold developers accountable. The two agendas are mutually dependent.

This paper proceeds as follows. Section 2 defines sociopolitical risks in more detail, distinguishes them from individual-level and existential risks, and clarifies their relationship to alignment. Section 3 presents six concrete failure modes located across institutional input, processing, and feedback channels. Section 4 outlines research and governance priorities, including directions for the ML community on system-level evaluation, alignment for epistemic health, and architectural constraints for governance-adjacent deployment. Section 5 considers alternative views.

## 2. Scope and Definitions

In this section, we formally define sociopolitical AI risks. Then, we distinguish sociopolitical risks from other risk

categories like individual-level harms and existential risks.

### 2.1. Working definition

Sociopolitical AI risks are threats to collective self-determination: a society's capacity to form shared judgments about common problems and to act on those judgments through accountable democratic institutions. This capacity rests on two interdependent pillars. The first is *social*: citizens and groups must be able to form, contest, and revise beliefs and preferences in ways that are not systematically distorted. The second is *political*: institutions must be able to register those signals, transform them into decisions, and be held accountable for the results. A risk is *sociopolitical* when it degrades either pillar, or weakens the connection between them.

These risks emerge when general-purpose AI (GPAI) is integrated into society at scales, speeds, and levels of opacity that institutions are not built for. Their relationship to alignment is not uniform. Some risks are *alignment-independent*, driven by cost and scale, and persist even when individual outputs are harmless (Sections 3.3, 3.4). Others are *alignment-caused*, in that reward shaping, safety filtering, and personalization can themselves produce side effects such as opinion flattening, sycophantic belief reinforcement, and the embedding of developer values into public infrastructure (Sections 3.1, 3.2, 3.6). A third category is *alignment-relevant*, where progress in interpretability and faithful reasoning could partially address institutional problems such as opaque decision-making (Section 3.5). This taxonomy suggests that model-level alignment and institutional resilience are complementary rather than competing.

The unit of analysis throughout is institutional rather than individual, as sociopolitical risks are system-level failures generated by aggregate deployment patterns rather than in-

dividual outputs. To make these dynamics tractable, we conceptualize institutions as information-processing systems operating across three stages, *input*, *processing*, and *feedback* (Figure 1)(Easton, 1965; Deutsch, 1963):

- **Input**: citizens send demands, preferences, and information through channels such as voting, public comment, litigation, and petition.

- **Processing**: institutions weigh and decide on these inputs through legislative deliberation, judicial reasoning, regulatory analysis, and administrative casework.

- **Feedback**: institutions communicate results through rulings, policy announcements, and enforcement actions, allowing citizens and oversight bodies to interpret outcomes, contest them, and inform future participation (Landemore, 2021).

This framework makes it possible to locate where AI disrupts institutional function and to distinguish failure modes that are otherwise hard to separate. The baseline, however, is not a well-functioning democracy, as existing participation channels already favor well-organized groups and policy outcomes already track wealth and influence (Gilens & Page, 2014). Our concern is not that AI degrades an idealized democracy but that it intensifies these inequalities and weakens the processes that make reform possible.

## 2.2. Relationship to adjacent risk categories and related work

Sociopolitical risks are related to, though analytically distinguishable from, two well-established categories of AI risk, and complementary to a third body of work on AI governance and sociotechnical systems.

**Individual-level harms.** Harms such as harassment, fraud, discrimination, and privacy violations remain important and are typically addressed through content moderation, access controls, or case-by-case enforcement (Weidinger et al., 2022). Sociopolitical risks arise when such harms scale or coordinate in ways that degrade institutions, as when persuasive content accumulates to shape electoral outcomes. Model-level safeguards reduce the incidence of these individual harms but do not by themselves address the aggregate institutional effects that emerge at scale.

**Existential risks.** One line of existential-risk work concerns scenarios in which a misaligned AI pursues goals incompatible with human survival (Bostrom, 2014; Russell, 2019) or in which catastrophic misuse produces irreversible harm (Sandbrink, 2023). A second line concerns the gradual erosion of human collective agency through incremental AI automation across economic, cultural, and governance domains (Kulveit et al., 2025; Drago & Laine, 2025). Our analysis connects to both by identifying specific institutional

mechanisms within the gradual-disempowerment arc, and by noting that when the institutions responsible for governing AI are themselves weakened, society loses the capacity to coordinate against catastrophic risks.

**Systemic risks and sociotechnical scholarship.** Taxonomies of societal-scale AI risk recognize that aggregate effects from many independently deployed systems form a category of their own (Critch & Russell, 2023), and Acemoglu (2024) describes a related feedback loop in which AI-driven worker displacement also erodes the political pressure workers could apply to address it. Sociotechnical scholarship argues that AI risks cannot be assessed at the model level alone, as they emerge through human interaction, organizational practice, and broader social systems (Weidinger et al., 2023; Dobbe & Wolters, 2024; Shelby et al., 2023). Platform governance research has further documented how automated content production shapes public discourse (Woolley & Howard, 2018) and how algorithmic monoculture produces correlated failures and homogenized outcomes across independent deployments (Kleinberg & Raghavan, 2021; Bommasani et al., 2022).

Where these traditions operate at the level of systems or macro-arcs, we provide a mechanism-level account of how AI degrades specific institutional processes, locating each failure in the governance loop and assessing what alignment can and cannot do about it. These mechanisms are already present with current AI capabilities and intensify as systems become more capable and widely deployed. Sociopolitical risks therefore complement existing model-level safety work and AI governance frameworks.

# 3. Sociopolitical Failure Modes

We organize representative failure modes by where they disrupt the governance feedback loop (Figure 1), describing for each the affected institutional context, the mechanism through which AI alters the dynamics, and the conditions under which the failure emerges.

## 3.1. Belief Homogenization

Existing AI models often produce lower output variance than the data they were trained on (Shumailov et al., 2024; Dohmatob et al., 2025; Wu et al., 2025a). Post-training methods such as reinforcement learning from human feedback (RLHF) and safety fine-tuning amplify this effect by systematically suppressing outputs that score poorly on helpfulness and safety criteria, narrowing the effective output distribution toward responses that conform to common rater preferences and safety policies (Ouyang et al., 2022; Bai et al., 2022a; Weidinger et al., 2021). Padmakumar & He (2024) confirm this empirically: while LLMs can imitate diverse styles, the semantic entropy of their outputs remains

lower than human baselines. Some convergence is beneficial, since consistency and mutual intelligibility have real value, yet the concern arises when convergence systematically suppresses alternative framings and problem definitions that models are otherwise capable of expressing.

As these systems are increasingly used in public-facing communication and policy work, their output priors enter public discourse at scale. Media effects research has long shown that influence operates less through direct persuasion than through *agenda-setting* (McCOMBS & SHAW, 1972) and *framing*, shaping which issues receive attention and how they are described (Entman, 1993). GPAI operates through these same indirect channels, and the evidence suggests its biases are not politically neutral. Motoki et al. (2024) find that LLM outputs favor certain political orientations while sidelining non-mainstream arguments, Buyl et al. (2026) show that models reflect the ideological commitments of their creators, and Jakesch et al. (2023) demonstrate experimentally that co-writing with an opinionated model shifts users' views, suggesting these biases reshape what people believe rather than merely coloring the produced text.

This compression matters because democratic deliberation depends on cognitive diversity (Landemore, 2021; Hong & Page, 2004). As users increasingly rely on AI to form positions rather than just express them (Lee et al., 2025; Chatterji et al., 2025), correlated model outputs shape the reasoning process itself, and public agreement can shift from being driven by shared evidence to being driven by shared model priors. It concentrates most acutely in deliberative and policy-making institutions (legislatures, courts, public commentary processes), where many independent actors draw on the same model priors, but also reaches everyday civic life, where AI increasingly shapes citizens' views on voting, commentary, and political conversation.

> **P1: Belief Homogenization**
>
> When many actors rely on a small number of similarly tuned models, the range of perspectives and arguments in public discussion narrows, and shared model priors increasingly shape what the public takes as common ground. This is also harder to detect than ordinary media influence, as AI shapes how people form opinions in private interactions where systematic bias has no public arena to be challenged. The risk concentrates in deliberative institutions and in risk-averse settings where institutions prefer consistency.

### 3.2. Belief Reinforcement

While **P1** concerns what perspectives are publicly available, **P2** concerns how an individual's beliefs change through sustained interaction with AI. Media reinforces existing beliefs

more than it converts them, one of the oldest findings in political communication (Lazarsfeld et al., 1968), reflecting a broader pattern of selective exposure in which people consistently seek out information that supports what they already believe (Klapper, 1960; Garrett, 2009; Waller & Anderson, 2021). Nguyen (2020) distinguishes echo chambers, where challenging views are actively excluded, from epistemic bubbles, where those views are simply absent. AI systems change several aspects of how both dynamics operate.

First, AI systems are *immersive*. Unlike broadcast media delivered as a one-way monologue, they engage in dialogue, adapting to pushback and supplying tailored rationales in real time, a form of interactive persuasion that more closely resembles the interpersonal influence Lazarsfeld et al. (1968) found to be more powerful than media exposure (Matz et al., 2024). Second, AI systems are *personalized at the individual level*. Where broadcast media targets demographics, AI can target an individual's specific doubts and reasoning patterns; Karadal & Kekulluoglu (2025) show that memory features allow chatbots to infer political orientation and shift responses accordingly, and when users signal belief in misinformation, model factual accuracy drops as the system pivots to validate the false premise (Jin et al., 2024; Piedrahita et al., 2026; Yadav et al., 2025a). Third, AI systems *accumulate*. As assistants adopt long-term memory, accommodation deepens across sessions rather than resetting, so interactions converge toward agreement instead of being driven by independent evidence (Perdomo et al., 2020; Hardt & Mendler-Dünner, 2025).

Systems built around behavioral prediction and retention are structurally rewarded for reducing friction (Zuboff, 2019), and RLHF reinforces this by optimizing for what an individual user perceives as helpful rather than what corrects them (Shapira et al., 2026; Turner & Eisikovits, 2026), producing sycophancy when there is no clear right answer (Sharma et al., 2024; Gabriel et al., 2024). The April 2025 GPT-4o rollback illustrates both the tendency and its limits: after the model became noticeably more flattering and agreeable, OpenAI rolled back the update, but the underlying training incentives persist (OpenAI, 2025a;d). The risk is most acute when individual users rely on a single AI assistant across many topics, persistent memory accumulates across conversations, and provider incentives favor retention over correction, each condition increasingly present.

Democracy depends on people being able to change their minds in light of new arguments (Landemore, 2021), and on justifications for political positions being open where others can challenge them (Habermas, 1996). AI-mediated reinforcement undermines both: it is built to please users rather than challenge them, and it happens in private where no one else can see the exchange or push back against it.

**P2: Belief Reinforcement**

AI assistants combine interactive personalization, long-term memory, and optimization for user satisfaction in ways that create private reinforcement loops, making users' beliefs progressively harder to revise. Because this reinforcement happens in private, builds up over time, and is driven by what retains users rather than what improves accuracy, it undermines the ability to change one's mind that democratic governance depends on. The risk takes its strongest form for individual citizens whose AI use replaces search and accumulates through long-term memory.

### 3.3. Congested Bureaucracy

After incorporating public input, institutions must process that input. In doing so, they rely on a limiting factor: **friction**. Writing a public comment, filing an appeal, or submitting a records request takes time and effort, and that friction acts as an implicit filter, keeping participation within the bounds of what human staff can read and decide on (Stephenson, 2006). This equilibrium is fragile even without advanced tools: after the 2020 U.S. election, organized activists flooded local offices with public-records requests challenging election results, diverting substantial staff time from core election administration (Layne, 2022; Green, 2024), showing that even coordinated human effort can tip the balance between generating and processing submissions until the filter breaks down.

General-purpose AI collapses this cost asymmetry at scale. A single actor can now generate large volumes of unique, plausible submissions (comments, appeals, complaints) at near-zero marginal cost, and agencies are often legally required to accept and process all of them (Levin, 2024). The challenge extends beyond volume to plausibility: LLM-generated content is comparable to human writing in argumentative settings (Herbold et al., 2023; Durak et al., 2025; Rathi et al., 2025), and reliably separating synthetic from genuine submissions at scale is difficult. Watermarking and detection methods exist (Wu et al., 2025b; Yang et al., 2025; Dathathri et al., 2024), but performance degrades under paraphrasing and light editing (Sadasivan et al., 2025; Lau & Zubiaga, 2025), and human reviewers are often swayed by writing style when factual grounding is weak (Fiedler & Döpke, 2025). The result is a congestion game (Rosenthal, 1973) over a shared resource (staff attention): it is individually rational to submit more, but the aggregate effect is a slower, more contested channel for everyone.

Faced with these dynamics, institutions have two main responses, both of which create new problems. They can deploy AI triage to manage volume, but the triage system then becomes a high-stakes filter whose errors and incentives shape who gets heard, requiring another layer of oversight to assess. A biased triage system can amplify some voices while dampening others. Alternatively, institutions can raise the cost of submission through more complex formats, paid or priority channels, or rules that narrow the agency's duty to respond. These measures reduce volume but shift representation toward actors who can more easily clear bureaucratic hurdles. Either path produces unequal access.

This failure mode persists regardless of alignment quality. Submissions need not be deceptive, biased, or policy-violating to overwhelm finite processing capacity; plausible, policy-compliant volumes can still be paralyzing. Model-centric evaluations cannot detect this because they assess individual outputs in isolation, while the binding constraint is aggregate load on institutional channels.

**P3: Congested Bureaucracy**

By making it cheap to generate large volumes of plausible civic submissions, AI can overwhelm administrative channels such as regulatory comment systems and benefits agencies. This failure mode arises when low generation costs combine with legal duties requiring institutions to accept all input and the absence of robust ways to authenticate participation without restricting access. Where these conditions hold, agencies will be pushed toward restrictive gating, with disproportionate impact on citizens who lack the resources to navigate added friction.

### 3.4. Epistemic Flood

Producing plausible text, audio, images, and video once required skilled labor or specialized tools. General-purpose AI now generates all of these at near-zero marginal cost. Verification still requires human judgment, source-checking, and institutional time. The asymmetry degrades shared reality. Establishing what happened, who said what, and what is real becomes systematically harder.

In March 2022, a deepfake video of Ukrainian President Zelenskyy calling for surrender circulated during the early days of the invasion. Even though the artifact was low-quality and quickly debunked, its timing forced the Ukrainian government to divert critical attention toward rapid refutation (AI Incident Database, 2022). The Biden robocall described in the introduction shares the same dynamic, being cheap to generate but expensive to investigate. Organized social-media manipulation is already a global phenomenon documented across dozens of countries (Bradshaw & Howard, 2019), and generative AI lowers the cost of producing the synthetic content such campaigns depend on.

Social media platforms allocate visibility through automated ranking signals like engagement, velocity, and repetition

across accounts. These signals are imperfect proxies for accuracy, and AI-generated content makes them substantially easier to manipulate at scale. A randomized experiment on Twitter found that algorithmic ranking amplifies political content relative to a chronological feed (Huszár et al., 2022), showing that this amplification dynamic operates in production systems. The result is that correction becomes expensive: rebutting a claim requires finding many variants, attributing and verifying provenance, producing counter-messaging, and distributing it through the same crowded channels fast enough to matter.

Existing safety evaluation frameworks provide an incomplete account of this threat. Model-level alignment helps at the margin, as a model that refuses deceptive impersonations prevents the most blatant attacks, but the verification asymmetry at the core of this failure mode is alignment-independent. High volumes of plausible, policy-compliant material impose the same verification burden as overtly deceptive content. The deeper risk is systemic, arising when content arrives faster than journalists, platforms, and institutions can verify, contextualize, and correct. Under those conditions, even when ground truth is recoverable in principle, establishing it as *common knowledge* in time to guide collective action becomes increasingly difficult.

---

**P4: Epistemic Flood**

When generating plausible political content becomes easier than verifying and correcting it, the binding constraint becomes verification and distribution bandwidth. This failure mode arises when a verification gap between cheap generation and slow human checking combines with public information environments whose visibility is allocated by manipulable ranking signals and time-sensitive decisions that require shared understanding of facts. The risk concentrates in institutions that depend on rapid, high-stakes verification, including election administrators, fact-checking organizations, journalists, and rapid-response oversight bodies.

---

### 3.5. Unauditable Authority

Modern governance rests on what Max Weber called rational-legal authority (Weber, 1978), the principle that state power must follow from public, intelligible rules rather than arbitrary will. This carries an implicit contract: coercive decisions are legitimate only insofar as affected parties can inspect, contest, and review the reasons behind them. A parallel logic governs private corporations, which operate within legal frameworks that presume regulators and courts can reconstruct how consequential decisions were made. We call this property auditability.

General-purpose AI threatens both sides of this relationship.

When decisions are mediated by systems whose reasoning cannot be reliably reconstructed, the machinery of oversight, including appeals, audits, investigations, and litigation, loses its force. This concern predates general-purpose AI, as earlier algorithmic systems in credit scoring, hiring, and policing already weakened accountability by hiding decisions behind proprietary processes (Pasquale, 2015). What makes general-purpose AI different is a combination of three factors that together overwhelm the mechanisms designed to hold decision-makers accountable.

First, **AI explanations cannot be reliably verified**. AI systems can produce after-the-fact explanations including chain-of-thought traces, but growing evidence suggests these do not reliably reflect how the model reached its output (Arcuschin et al., 2025), and an explanation that cannot be trusted cannot ground a meaningful appeal (Vredenburgh, 2022). Second, **scale defeats case-by-case oversight**, as a human caseworker handles hundreds of decisions while an AI system processes millions, and automating the oversight itself reintroduces the problem one level up, as with the AI triage systems in Section 3.3. Third, **institutional access barriers compound technical opacity**, as a proprietary model's weights and training data may be shielded by trade secrets or contractual confidentiality, letting opacity serve as a deliberate shield against enforcement rather than an incidental byproduct of complexity.

Each factor alone might be manageable, but together they create an accountability gap that existing oversight was not built to handle. Citizens cannot contest decisions they cannot examine, and regulators face a higher bar when they cannot reconstruct how a decision was made. Progress on interpretability and chain-of-thought faithfulness would directly address the first factor, making it where alignment research and institutional accountability reinforce each other, but the second and third require institutional and legal responses that technical advances alone cannot provide.

---

**P5: Unauditable Authority**

AI opacity, whether technical or institutional, erodes accountability in two directions. Citizens and oversight bodies lose the ability to audit government decisions, while regulators lose the ability to investigate corporate conduct. This failure does not stem from AI being a simple "black box," but arises when three conditions combine: unverifiable explanations, decision volumes that defeat case-by-case review, and access barriers that block independent inspection. The risk concentrates wherever consequential decisions are delegated to AI (courts, regulatory agencies, benefits agencies), and wherever firms deploy opaque systems that shield their conduct from regulatory investigation.

---

### 3.6. Normative Centralization

States can be coerced through infrastructural choke points. The SWIFT financial messaging network and the U.S.-controlled Global Positioning System (GPS) show how strong network effects and concentrated control over a critical interface let one state exert power over others through the threat of exclusion rather than direct force (Farrell & Newman, 2019). Frontier AI is converging on a similar structure, but with several choke points instead of one. Most countries cannot train frontier models themselves, so their administrative and economic systems increasingly depend on AI capabilities they do not control, giving providers structural power over the governments that procure their models (Strange, 1996). Three choke points stand out, namely the compute supply chain, which is highly concentrated and can constrain who is able to train state-of-the-art systems (Sastry et al., 2024), cloud access, which concentrates the capacity to run these systems in a small number of providers that can deny or throttle service, and model behavior itself, shaped by a constitution or model spec that defines permissible use, acceptable topics, and value commitments (Bai et al., 2022b; Anthropic, 2026b; OpenAI, 2025c).

Compute and cloud follow a familiar logic in which access can be denied, throttled, or made conditional, and the affected state knows it is being coerced, as semiconductor export controls illustrate. The third choke point, the model's own normative constraints, is harder to see. Because a model's value commitments are built into the capability itself, their influence is gradual and continuous, and may not register as coercion at all. The developer sets the constitution unilaterally, determining what policy advice the AI gives, which positions it treats as legitimate, and which requests it refuses. When these defaults reflect the developer's home jurisdiction, governments that adopt the model import those commitments into their own administration, shifting normative authority from elected officials toward model providers. This resembles what Lazar calls the "intermediary power" platforms hold over the social relations they mediate (Lazar, 2025), though its novelty here lies in operating from developer to procuring government, where the stakes are sovereignty and democratic accountability rather than individual autonomy.

This normative influence extends even to the developer's home government. In the ongoing dispute between Anthropic and the U.S. Department of Defense, the company's use policy itself is being contested, restricting applications such as autonomous weapons and domestic surveillance. Where an ordinary procurement fight hinges on price, delivery, or specifications, this one hinges on the model's permitted normative behavior, and even the government procuring the system can contest those defaults only through contract leverage and litigation (Sayler, 2026). What makes this threat model distinctive is that it does not require an alignment failure: the model behaves exactly as its constitution specifies, and that is precisely the problem for governments whose values differ from the developer's. P6 is therefore the clearest case of the alignment-caused category from Section 2.1.

Open-weight models offer a partial counterweight, as a state with enough compute can adapt one to encode its own values. But few states can. Compute hardware remains concentrated, and frontier providers offer integration and support that adapted alternatives lack, so procurement tends to default to off-the-shelf models. A coalition can pool compute to clear that barrier, but then inherits a smaller version of the same problem, as it must agree internally on whose values the shared model encodes. The risk concentrates, then, among states without the means to adapt a model alone or to shape what a shared one encodes.

---

**P6: Normative Centralization**

Unlike traditional choke points that operate through access denial, frontier AI carries normative constraints built into constitutions, model specs, and usage policies. The conditions for this failure mode are: a government depends on frontier AI it cannot independently train or adapt, the model carries developer-set normative constraints, and the procuring state lacks processes to customize or contest them. Where these hold, normative authority shifts from elected officials to a small group of constitution designers, weakening sovereignty for procuring states and bypassing democratic accountability even within the developer's home jurisdiction.

---

## 4. Recommendations

**R1: Develop institution-specific threat models and safety thresholds for sociopolitical risks.** Each major institution (legislatures, courts, regulatory agencies, electoral systems) should formalize threat models that identify how AI alters their input, processing, and feedback layers, and specify capability thresholds at which risks emerge and cascade (P3, P4, P5). We propose encoding these thresholds as Institutional Safety Levels (ISLs) for public-sector AI deployment, analogous to capability-triggered frameworks that frontier AI developers have adopted for model-level risks (Anthropic, 2026a; Google DeepMind, 2025; OpenAI, 2025b). Each ISL *binds* concrete AI capabilities to mandatory procedural safeguards, rather than merely reporting risks after the fact. For example, in a court system, the shift from AI drafts internal research memos" to AI generates sentencing or bail recommendations" would automatically trigger disclosure to affected parties, retention of full reasoning traces, and mandatory human sign-off with appeal

pathways; higher-impact tiers, such as drafting legally operative text or enabling population-scale filing, would require external audit or pre-deployment authorization.

Thresholds should be set through democratic input processes rather than determined unilaterally by technical experts (Vaintrob, 2025). Since reliably measuring AI capabilities remains an open challenge, ISLs should not depend on technical measurement alone and should default to requiring procedural safeguards when measurement is uncertain; enforcement mechanisms for these requirements warrant further specification. This creates a forward-looking regime in which institutions pre-commit to governance actions as AI capability scaling triggers legitimacy- and trust-relevant thresholds, particularly in settings where adoption gaps between regulated actors and institutions can destabilize institutional equilibria (Vaintrob, 2025).

**R2: Expand AI safety evaluations beyond model-level harms to include sociopolitical effects.** The failure modes we identify require benchmarks that assess how individually benign outputs aggregate into systemic effects on institutions and public discourse (**P1**, **P2**, **P3**, **P4**) (Yadav et al., 2025b; Pandey et al., 2026), a gap that existing model-level evaluations do not fill (Costello et al., 2024; Salvi et al., 2025). For sociopolitical effects (**P1**, **P2**), benchmarks should measure outcomes such as opinion diversity, narrative convergence, polarization, and epistemic drift; for saturation risks (**P3**, **P4**), participatory channels should be stress-tested as systems under load, with metrics for throughput collapse, human displacement, and error amplification. We propose multi-agent simulation as a tractable first step: agent-based modeling has established precedent in computational social science (Epstein & Axtell, 1996), and recent work shows LLM-powered agents can replicate human behavior in social experiments (Park et al., 2023; Argyle et al., 2023; Aher et al., 2023) and simulate legislative processes (Baker & Azher, 2024).

Concrete evaluation questions include: at what submission volume does a comment system's signal-to-noise ratio collapse, how does opinion diversity shift when a majority of simulated participants use the same foundation model, and how does the cost of correcting a false belief scale with the duration of sycophantic reinforcement? We frame simulation as a complement to population-level empirical studies, not a substitute; LLM-based simulation inherits biases from the models it uses, and recent work suggests that LLM agents may lack reliable awareness of each other as interlocutors (Choi et al., 2025), introducing additional distortions when simulating multi-party deliberation. Results therefore require calibration against real-world baselines, and population-level empirical validation of how widespread LLM use reshapes group reasoning and the diversity of public argumentation remains an important open challenge.

**R3: Increasing trust and robustness in deployed AI systems.** Institutional AI systems should log governance-grade decision records by default to address the accountability and provenance failures that characterize **P3** and **P5**: durable, standardized traces that capture inputs, model and prompt versions, tool calls, retrieved sources, intermediate state, and uncertainty, in formats suitable for audit, comparison, and legal review (Mitchell et al., 2019; Raji et al., 2020). These records should be queryable across cases to enable systematic auditing, adversarial stress-testing, and cross-institutional comparison of decision patterns, building on emerging standards for AI system logging (International Organization for Standardization, 2026; UK Government Digital Service, 2025).

Explanations need to move beyond chain-of-thought, which can be unreliable (Arcuschin et al., 2025), toward methods that make input-output dependencies explicit and stable under perturbation, enabling officials to understand how recommendations change when conditions change; this includes improving the faithfulness of reasoning traces (Yu et al., 2026) and causal and counterfactual explanation methods (Wachter et al., 2018). Separately, deployed systems need to track provenance and support proof-of-personhood for inputs such as public comments, filings, and petitions, so institutions can distinguish genuine participation from automated volume without compromising privacy. Recent work proposes privacy-preserving personhood credentials using verifiable credentials and zero-knowledge proofs that avoid the centralization risks of biometric systems (Adler et al., 2025), though practical tradeoffs between verification strength and privacy remain an open design challenge. Together, these measures make AI-mediated governance inspectable, contestable, and reliable at scale.

**R4: Enable pluralistic alignment in public AI systems.** When a single model family is deployed across public institutions, its training data, safety filters, and reward functions effectively standardize how arguments are framed and which claims are treated as legitimate, producing epistemic monoculture and normative centralization (**P1**, **P2**, **P6**) (Gabriel & Keeling, 2025). This risk is compounded by the fact that procurement is increasingly the de facto site of AI governance, where consequential normative decisions are made through bilateral vendor negotiations without public accountability (Alais, 2026). We therefore reframe pluralistic alignment (Sorensen et al., 2024) as a procurement diversification requirement: public procurement frameworks should mandate interoperability standards and data portability so that institutions are not locked into a single provider, and should support multi-provider deployment strategies that enable switching providers without re-engineering workflows.

Deployments should expose common interfaces for reasoning traces and decision logs, enabling systematic compar-

ison of outputs on identical inputs; cross-model disagreement checks and periodic re-benchmarking can surface blind spots that arise when similar training produces similar biases. Decision-log retention and explainability should be mandatory procurement requirements, modeled after data retention standards in regulated sectors such as finance, so that model behavior and normative trade-offs remain auditable over time. This framework applies most directly to institutional procurement and does not resolve AI capabilities embedded within specific commercial applications, such as a text editor with built-in AI, which we flag as an important boundary condition; implementation costs and disagreement resolution across providers remain open research priorities.

## 5. Alternative Views

**Societies will gradually adapt without intervention.** From a Hayekian perspective, complex social systems reach equilibrium through decentralized self-adaptation rather than centralized design (Hayek, 2013; Scott, 1998). Some argue that AI-induced disruptions will be absorbed through evolving social norms, market incentives, and trust heuristics (Folke et al., 2005), and that proactive intervention may overestimate our ability to anticipate and manage such systems. Historical precedent offers genuine support, as institutions adapted to the printing press and the internet by developing new procedures and oversight mechanisms over time (Wu, 2010; Kissinger et al., 2021), and decentralized adaptation may outperform centralized design in complex systems.

However, the current situation may be structurally different: the adaptation this view assumes may itself be undermined by the dynamics it expects to resolve. Acemoglu (2024) argues that AI development concentrating economic rents also erodes the political and organizational capacity on which institutional self-correction depends. If this reinforcing dynamic holds, the capacity for institutional response is degraded precisely when it is most needed — not only because the pace of change is unprecedented, but because the adaptive machinery that resolved prior transitions may no longer be intact.

**Sufficient alignment will prevent sociopolitical risks.** This view holds that advances in technical alignment will mitigate sociopolitical AI risks at the model level, reducing the need to analyze institutional dynamics (Russell, 2019; Ouyang et al., 2022; Bai et al., 2022b). We agree that alignment is necessary and can directly help with specific risks: less sycophantic training reduces belief reinforcement (Sharma et al., 2024), improved chain-of-thought faithfulness strengthens institutional auditability (Yu et al., 2026), and models that refuse to generate deceptive content limit the most blatant epistemic attacks.

However, the relationship between alignment and sociopolitical risks is more complex than this view suggests. Some failure modes persist regardless of alignment quality: a perfectly aligned model still makes it cheap to overwhelm a public comment system (P3), and alignment does nothing to prevent the displacement of human participation that weakens citizen leverage over institutions. Others are direct consequences of how current alignment methods work: the same reward signals that make models safe and helpful also narrow output diversity (P1) and embed developer values into public infrastructure (P6) (Gabriel & Keeling, 2025). Moreover, whether alignment delivers its sociopolitical benefits depends on institutional conditions: who sets the alignment objectives, how compliance is verified, and whether deployment contexts negate model-level gains. Alignment without functioning oversight institutions risks optimizing for the wrong objectives; institutional resilience without alignment lacks the technical tools to govern model behavior. The two agendas are mutually dependent.

## 6. The Way Forward

In this position paper, we argue that sociopolitical AI risks emerge at the level of institutions and governance systems and cannot be resolved by model-level alignment alone. Advancing this agenda requires coordinated action across multiple communities. For the AI research community, this means extending safety work toward system-level evaluations that capture aggregation effects, institutional load, and belief dynamics under realistic deployment conditions. For AI developers, it entails treating contestability, auditability, and pluralism as core design principles when integrating AI into public-facing systems, rather than post-hoc remedies. In parallel, policymakers must move beyond reactive controls toward institution-level safeguards that preserve democratic responsiveness at scale. We call on all three communities to ensure that rapidly advancing AI is matched by corresponding progress in institutional resilience.

## Acknowledgments

The authors thank Ashton Anderson, Brenda Baker, Yoshua Bengio, Matthias Bethge, Pepijn Cobben, Giulio Corsi, Eric Grosse, Roger Grosse, Karoline Helbig, Xuanqiang Angelo Huang, Aidan Kierans, David Lie, Marcelo Sartori Locatelli, Suvajit Majumder, Richard Mallah, Susan Nesbitt, Susan Perry, Paul Resnick, Stuart Russell, Mrinmaya Sachan, Stephan Schwahlen, and Audrey Tang for helpful feedback and discussions on the paper. This work was supported in part by the German Federal Ministry of Education and Research (BMBF): Tübingen AI Center, FKZ: 01IS18039B; by Coefficient Giving; and by the Machine Learning Cluster of Excellence, EXC number 2064/1 – Project number 390727645.

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
