# OpenReview forum: "Position: Safe Models Do Not Guarantee Safe Societies"
_ICML.cc/2026/Position_Paper_Track — ICML 2026 Position Paper Track spotlight_

### Official Review · Reviewer_hGZM · 2026-03-06

**Significance:** 1
**Argument Clarity:** 2
**Rating:** 4
**Confidence:** 4

**Questions:**

1. How do you conceive of the relation between *sociopolitical* risks and *sociotechnical* risks from AI?
2. What conditions need to be met for each of the failure modes to be of particular concern?

**Alternative Views Section:**

Yes

**Compliance With Llm Reviewing Policy A Conservative:**

Affirmed.

**Discussion Potential:**

2

**Final Justification:**

The authors' rebuttal was strong, and, assuming that revisions are high-quality, I feel comfortable that this paper can be marginally above the acceptance threshold.

**Paper Summary:**

This paper introduces the notion of *‘sociopolitical AI risks’*, and presents 6 *‘failure modes’* in which such risks may arise within the sociopolitical process. It argues that model-level safeguards, as usually discussed in relation to AI alignment, are insufficient for dealing with such risks.

**Position:**

Yes

**Position In Title:**

Yes

**Related Work:**

1

**Strengths And Weaknesses:**

**Strengths:**
- The overall framing and breakdown of the 6 failure modes is instructive, and situating them in sociopolitical processes is helpful.
- The paper is structured and written clearly.
- The topic is relevant, timely, and worthy of investigation.

**Main weaknesses:**
- While the position is stated clearly in the introduction, the specific contribution being made to the discourse feels fairly thin -- essentially boiling down to introducing a 'distinct class' of risk. While the second half of the position (that model-level alignment frameworks cannot address this class of risk) is a more substantial position, very limited space is given to arguing for this in the paper. In particular, no argument is made regarding the urgency/magnitude of these risks. As such, I’m left feeling unconvinced that this is something that I should really care about.
- Secondly, I find that there is insufficient discussion of existing literature, and many unsupported claims. The paper does not mention the concept of *sociotechnical* framings of AI impacts and risks, despite the concept of *sociopolitical AI risk* having many similarities. (On sociotechnical approaches, see e.g., https://datasociety.net/wp-content/uploads/2024/05/DS_Sociotechnical-Approach_to_AI_Policy.pdf; https://arxiv.org/abs/2310.11986; https://dl.acm.org/doi/full/10.1145/3706599.3706747; https://link.springer.com/article/10.1007/s11023-024-09668-y). This lack of interaction with closely-related literature i) means that the paper's contribution does not feel novel (compared to e.g. a paper that situates sociopolitical risk within the broader sociotechnical framing of AI), and ii) makes the comparison to ‘model-level alignment’ feel like a straw-man. Additionally, many key claims made are stated without evidence or argument, undermining the strength of the paper. Examples include: claims regarding choke points in the AI supply chain (see e.g. https://arxiv.org/pdf/2402.08797); the claim in Section 3.4 that *"Existing research frameworks provide an incomplete account of this threat."* (which existing research frameworks are being referred to?); and the claim (also in Section 3.4) that *"creating content scales easily; checking it does not"*.

An additional minor weakness is the considerable amount of (almost verbatim) repetition between sections. E.g. the discussions of watermarking in sections 3.3 and 3.4; and the repeated story of spoof Biden phone calls in the Introduction and section 3.4.

**Support:**

2

---

> ### Author Rebuttal · Authors · 2026-03-31
>
> We thank the reviewer for their detailed engagement, particularly the call to situate our contribution relative to the sociotechnical literature. We address each concern below.
>
> **On the relationship between sociopolitical and sociotechnical risks.** The sociotechnical literature the reviewer cites (Weidinger et al., 2023; Dobbe & Wolters, 2024; Chen & Metcalf, 2024) asks whether an AI system is safe in the context where it is deployed, with an individual ML application as the unit of analysis. We share this tradition's core observation that model-level evaluation is insufficient. Our contribution addresses a different question: what happens when many individually safe-in-context AI systems operate simultaneously at scale? The unit of analysis shifts from the application to the institution. In the revision, we will make this relationship explicit and cite Weidinger et al. (2023), Dobbe & Wolters (2024), Chen & Metcalf (2024), and Shelby et al. (2023).
>
> **On substance and novelty.** We recognize that the submitted version may have understated what the paper actually argues. The position is not simply that sociopolitical risks exist, but that they have a specific relationship to alignment that matters for how the ML safety community allocates effort. In the revision, we make this explicit: some sociopolitical risks are alignment-independent (they persist regardless of model-level safety: P3, P4), others are alignment-caused (they arise directly from current alignment methods such as RLHF reward shaping and constitutional constraints: P1, P2, P6), and others involve alignment-relevant limitations that compound (P5). This communicates to researchers which risks their current tools can address, which risks require institutional along with technical interventions, and which risks involve design tradeoffs. We believe this is a substantive claim that goes beyond surveying known problems.
>
> **On urgency.** The reviewer notes that no argument is made for why these risks warrant priority. We will strengthen this in two ways. First, we make the mutual-dependence argument more forcefully: the institutions responsible for governing AI (regulatory agencies, courts, legislatures) are themselves targets of the failure modes we describe, creating a feedback loop where governance capacity degrades precisely as governance needs to intensify. Second, we will highlight concrete evidence of these dynamics already materializing, e.g., the GPT-4o sycophancy incident and rollback (OpenAI, 2025a,b), and documented patterns of computational propaganda campaigns across dozens of countries (Woolley & Howard, 2018).
>
> **On unsupported claims.** We take this seriously and will address the specific examples the reviewer flags.
>
> On choke points in the AI supply chain: in the revision, we will cite Meiklejohn et al. (2025) on ML supply chain risks for open-weight models and Casper et al. (2025) on concentration in compute and data infrastructure as concrete evidence for the structural dependencies we describe.
>
> On "existing research frameworks provide an incomplete account": the reviewer is right that this claim needs a referent. We are referring to the misinformation detection literature, which evaluates whether individual synthetic artifacts can be detected and labeled, rather than assessing what happens when content arrives faster than verification infrastructure can process it. In the revision, we will make this explicit and cite Woolley & Howard (2018) and Howard (2020) on computational propaganda as evidence that the exploitation of verification asymmetries is already a documented global pattern.
>
> **On conditions for failure modes.** In the revision, we will specify what conditions must hold for each threat model. For example, belief reinforcement (P2) requires that AI assistants are replacing disagreement-exposing information sources, that long-term memory is widely adopted, and that provider incentives favor retention over correction.
>
> **New references**
>
> Casper, S., et al. Open technical problems in open-weight AI model risk management. arXiv, 2025.
>
> Chen, B. & Metcalf, J. A Sociotechnical Approach to AI Policy. Data & Society, 2024.
>
> Dobbe, R. & Wolters, A. Toward Sociotechnical AI: Mapping Vulnerabilities for Machine Learning in Context. Minds and Machines, 34(12), 2024.
>
> Howard, P. N. Lie Machines. Yale University Press, 2020.
>
> Meiklejohn, S., et al. Position: ML models have a supply chain problem. ICML Position Paper Track, 2025.
>
> Shelby, R., et al. Sociotechnical Harms of Algorithmic Systems. AIES '23, 2023.
>
> Weidinger, L., et al. Sociotechnical Safety Evaluation of Generative AI Systems. arXiv:2310.11986, 2023.
>
> Woolley, S. C. & Howard, P. N. Computational Propaganda. Oxford University Press, 2018.

---

> > ### Author Rebuttal · Reviewer_hGZM · 2026-04-03
> >
> > Thanks to the author(s) for their thorough response.
> >
> > Overall, I feel satisfied that my concerns have been addressed in the rebuttal. Though, as reviewer oAKF points out, some of the revisions are fairly substantial. Given that I am not able to review any changes directly, I have to assume that revisions are high quality.
> >
> > Given the above, I will raise my decision to 4) borderline accept.

---

### Official Review · Reviewer_oAkf · 2026-03-13

**Significance:** 1
**Argument Clarity:** 2
**Rating:** 4
**Confidence:** 4

**Questions:**

-In section 4 R1, the example mechanisms seem to mostly introduce reporting around the use of potentially problematic AI, but don't seem to prevent it or mitigate the harms (or at least such benefits are not apparent to me)

-For section 4 R2, what would such benchmarks look like? Assessing how a model will shift the public opinion distribution seems difficult other than via mass user studies or surveys before and after wide deployment, which seems both scientifically challenging to extract signal from (due to many possible confounding factors) and a bit pointless, since the damage has already been done if the effect can be seen at a population level. Further, who would maintain and administer such benchmarks? Is this a regulatory mechanism that gates market access, or an academic study used to inform policy with no binding authority in itself?

-For section 4 R3, calling for explainable representations that are robust to perturbation and not prone to error/hallucination is great, but this goal has proven difficult to date despite much research. What can we do about these issues in the absence of such a strong level of interpretability? Similarly, what does proof-of-personhood look like? This seems difficult to collect and validate in many domains without restricting input to in-person submission or committing gross privacy violations such as collecting and storing biometric data like face scans.

-For section 4 R4, what is the operative model here? Is this imagining a scenario where public-facing AI models are accessed through a government or independent third party-maintained universal interface, which uses multiple commercial models on the back end? How does this work with AI model deployments in diverse application domains or embedded within specific applications (for example, a text editor with built-in proofreading and editing suggestion AI)? I'm not sure how this concept is supposed to actually work in practice.

**Alternative Views Section:**

Yes

**Compliance With Llm Reviewing Policy A Conservative:**

Affirmed.

**Discussion Potential:**

1

**Final Justification:**

The authors have substantially revised the paper's position and framing, which is I think enough to address the core concerns I had. As this is a fairly substantial revision I'm not comfortable to endorse the work unconditionally, but I trust from the description of the changes that the revised manuscript addresses the critical issues.

**Paper Summary:**

This paper describes a category of AI risk, sociopolitical risk, which damages the ability of society and government feedback mechanisms to function at various stages in the process due to unique properties and abuse of modern AI models. The paper then describes a number of approaches to address socioeconomic risks of different types.

**Position:**

Yes

**Position In Title:**

No

**Related Work:**

3

**Strengths And Weaknesses:**

This paper collates a wide range of evidence and examples of AI harms and places them cleanly into a framework for how they can interfere with the mechanisms of governance, with well supported examples of these harms and clear links to proposed mechanisms to address them.

However, this paper has a few major issues:

Firstly, the "position" of the paper is not clearly presented. Most of the paper is focused on defining socioeconomic risk types and proposing mechanisms to address them, and it seems like the position is simply "socioeconomic risks exist" which seems uncontroversial as a position- most of the specific issues raised have been previously raised in one form or another, so it seems like the real contribution here is collating them into a single framework and proposing solutions to address them. This could be a good contribution, but it isn't framed as a position with supporting arguments and evidence (for example, it could be argued that there is value in treating these different hazards within a cohesive framework where root causes can be addressed). Because of this, I'm not sure how much potential for discussion or awareness-raising this paper has. All of the specific risks described are known to the AI policy/safety community, and placing them within a coherent framework is poorly motivated at present (but I am open to persuasion on the benefits of doing so).

Second, the writing style of the paper is very dry and prone to terminology spam, using very specific (and sometimes verbose) terms to describe every risk in rather clinical terms. While formal definitions are useful, a balance must be struck between correctness and clarity, and in my opinion the current paper struggles to communicate ideas succinctly and clearly, and excessive use of formal terminology distracts from arguments and evidence rather than supports it.

Third, the policy recommendations to address socioeconomic risks lack important details and seem poorly thought out. I go into more details in the questions below, but each one seems to have either major unaddressed challenges to practical implementation or is vague to the point of uselessness. The section describing the various socioeconomic risk types suffers somewhat from being vague and over-general, but does have examples and details to ground it, which the recommendations lack. I'm not sure these provide any value to the reader as-is.

Given the above, I'm inclined to recommend rejection. The ideas in this paper could make for a good position paper, but more work is needed to motivate the paper's framing and explore some of the implications/solutions in more detail.

**Support:**

3

---

> ### Author Rebuttal · Authors · 2026-03-31
>
> We thank the reviewer for their careful reading and specific questions about each recommendation. We address the main concerns below.
>
> **On the paper's position.** We appreciate this pushback and recognize that our framing may not have made the position's force sufficiently clear. Our claim is not simply that sociopolitical risks exist; we agree that would be uncontroversial. The position is that a specific and important subset of these risks is alignment-independent: they persist even if every model is perfectly aligned with its operator's intent and values. This has a concrete implication that we believe is underappreciated in the ML safety community: no amount of progress on RLHF, constitutional AI, or interpretability will resolve risks driven by cost asymmetries and deployment scale (P3, P4), and current alignment methods actively contribute to others (P1, P2, P6). The governance feedback loop framework serves as an analytical tool for identifying where each mechanism disrupts institutional function and why model-level fixes are insufficient at each point. We will sharpen this argumentative throughline in the revision.
>
> **On writing style.** We take this point seriously and will reduce terminological density in the revision, particularly in Section 3 where formal definitions can be streamlined without loss of precision.
>
> **On recommendations (R1-R4 in the submitted version).**
>
> *R1/ISLs: Do these prevent harm or merely report it?* ISLs bind concrete AI capabilities to mandatory safeguards. For example, the shift from "AI drafts internal research memos" to "AI generates sentencing recommendations" would automatically trigger disclosure to affected parties, retention of reasoning traces, mandatory human sign-off with appeal pathways, and (at higher-impact tiers) external audit or pre-deployment authorization. In the revision, we will add that the thresholds themselves should be set through democratic input processes rather than determined unilaterally by technical experts. We acknowledge that enforcement mechanisms require further specification and will discuss this more explicitly.
>
> *R2/Benchmarks: What would sociopolitical benchmarks look like?* In the revision, we propose multi-agent simulation as a concrete methodology. Agent-based modeling has established precedent in computational social science (Epstein & Axtell, 1996), and recent work shows LLM-powered agents can replicate human behavior in social experiments (Park et al., 2023; Aher et al., 2023). Concrete evaluation questions include: at what submission volume does a comment system's signal-to-noise ratio collapse? How does opinion diversity change when a majority of simulated participants use the same foundation model? We agree that these are not population-level field studies; we frame simulation as a tractable first step, with the acknowledged limitation that results require calibration against real-world baselines. We will also make clear that population-level empirical validation remains an open challenge.
>
> *R3/Proof-of-personhood: What does this look like practically?* In the revision, we will cite Adler et al. (2024), which proposes privacy-preserving personhood credentials using verifiable credentials and zero-knowledge proofs. These approaches avoid the centralization risks of biometric systems that the reviewer rightly flags as concerning. We will add a brief discussion of the practical tradeoffs.
>
> *R4/Pluralistic alignment: What is the operative model?* In the revision, we clarify this as a procurement diversification requirement rather than an interface. Public procurement frameworks should mandate interoperability standards and data portability so institutions are not locked into a single provider, and should support multi-provider deployment strategies that enable switching providers without re-engineering. We acknowledge that this applies most directly to institutional procurement contexts and does not resolve the case of AI capabilities embedded within specific commercial applications (e.g., a text editor with built-in AI), which we flag as an important boundary condition.
>
> **New references**
>
> Adler, S., Hitzig, Z., Jain, S., et al. Personhood Credentials: Artificial Intelligence and the Value of Privacy-Preserving Tools to Distinguish Who Is Real Online. arXiv:2408.07892, 2024.
>
> Aher, G., Arriaga, R. I., & Kalai, A. T. Using Large Language Models to Simulate Multiple Humans and Replicate Human Subject Studies. ICML, 2023.
>
> Epstein, J. M. & Axtell, R. Growing Artificial Societies: Social Science from the Bottom Up. Brookings Institution Press and MIT Press, 1996.
>
> Park, J. S., O'Brien, J. C., Cai, C. J., Morris, M. R., Liang, P., & Bernstein, M. S. Generative Agents: Interactive Simulacra of Human Behavior. UIST '23, 2023.

---

> > ### Author Rebuttal · Reviewer_oAkf · 2026-04-03
> >
> > Thanks for the response! I appreciate the changes of framing and context described, and while they are substantial enough that I would need to re-review the whole paper to remove all misgivings, based on this I feel confident enough to raise my score on the assumption that these revisions are well integrated into the paper and its framing. While I share some of reviewer hGZM's concerns regarding significance, I think I lean towards acceptance on the belief that these issues have been addressed somewhat by revisions.

---

### Official Review · Reviewer_EykN · 2026-03-13

**Significance:** 4
**Argument Clarity:** 4
**Rating:** 5
**Confidence:** 4

**Questions:**

In R3, you recommend logging full reasoning traces, tool calls, and intermediate states, but I think further discussion about how to use these datasets for increasing trust and robustness could be more helpful for the community.

**Alternative Views Section:**

Yes

**Compliance With Llm Reviewing Policy A Conservative:**

Affirmed.

**Discussion Potential:**

3

**Final Justification:**

The rebuttal addressed all my concerns, and I therefore maintain my score in favor of acceptance.

**Paper Summary:**

In this position paper, the authors define a distinct class of AI risks, i.e., sociopolitical risks, and claim that these risks cannot be adequately addressed by existing model-level alignment frameworks. These risks relate to a society’s capacity to articulate collective interests and realize them through accountable institutions. The central argument is that even if individual AI models are “safe” (i.e., aligned with human intent and values), their widespread integration into social and political systems may still degrade institutional functioning by disproportionately amplifying the scale, speed, and opacity of operations.

**Position:**

Yes

**Position In Title:**

Yes

**Related Work:**

4

**Strengths And Weaknesses:**

Strengths：

1. This paper clearly differentiates "sociopolitical risks" from "individual-level harms" (like bias) and "existential risks" (like loss of control), suggesting that these risks persist even if models are perfectly aligned with user intent.

2. The paper shifts the "AI Safety" unit of analysis from individual model outputs to systemic, institutional effects by framing governance as an information-processing loop.

Weaknesses：

1. In lines 165–167, the authors claim that agreement is often locally rewarded while disagreement is conversationally costly. It would be interesting to further discuss the potential role of AI in human deliberation and supporting negotiation between people.

**Support:**

4

---

> ### Author Rebuttal · Authors · 2026-03-31
>
> We thank the reviewer for their supportive and thoughtful review. We address both suggestions below.
>
> **On AI's potential role in deliberation and negotiation.** We agree this deserves more attention. In the revision, we expand on this in two places. First, in Section 3, we add an explicit counterargument noting that AI assistants could in principle reduce polarization: a system that understands a user's values could translate opposing arguments into personally compelling language, present balanced evidence, or flag uncertainty (Argyle et al., 2023). Second, we add a new recommendation on deliberative infrastructure, discussing structured deliberation platforms that surface consensus rather than amplify volume (Small et al., 2021), and citing evidence that AI-mediated deliberation can operate at scale: Tessler et al. (2024) showed that an LLM-based mediator generated group statements preferred over those from human mediators across thousands of participants. We also discuss Taiwan's 2024 citizen assembly on deepfake regulation as a concrete case where AI-supported deliberation produced effective legislation.
>
> **On how logging datasets can be used for trust and robustness.** The reviewer is right that describing what to log is insufficient without articulating how those records would be used. In the revision, we expand the recommendation to outline concrete uses: logged decision records should be queryable across cases to enable systematic auditing, adversarial stress-testing (probing how outputs change when inputs change) and cross-institutional comparison of decision patterns. We also connect this to the broader agenda in causal and mechanistic learning, which aims to make input-output dependencies explicit and stable under perturbation (Yu et al., 2025).
>
> **New references**
>
> Argyle, L. P., Bail, C. A., Busby, E. C., et al. Leveraging AI for Democratic Discourse: Chat Interventions Can Improve Online Political Conversations at Scale. Proceedings of the National Academy of Sciences, 120(41):e2311627120, 2023.
>
> Small, C., Bjorkegren, M., Erkkila, T., Shaw, L., & Megill, C. Polis: Scaling Deliberation by Mapping High Dimensional Opinion Spaces. Recerca: Revista de Pensament i Analisi, 26(2), 2021.
>
> Tessler, M. H., Bakker, M. A., Jarrett, D., et al. AI Can Help Humans Find Common Ground in Democratic Deliberation. Science, 386, 2024.

---

> > ### Author Rebuttal · Reviewer_EykN · 2026-04-02
> >
> > Based on the rebuttal and proposed revisions, I maintain my positive assessment and score for this submission.

---

### Official Review · Reviewer_Cw4V · 2026-03-24

**Significance:** 4
**Argument Clarity:** 4
**Rating:** 5
**Confidence:** 4

**Questions:**

Could the paper engage more with the related literature, as detailed above, especially democratic theory?

Could the paper strengthen the alternative perspectives section?

Could the paper make the recommendations more concrete and actionable for the ICML audience?

**Alternative Views Section:**

Yes

**Compliance With Llm Reviewing Policy A Conservative:**

Affirmed.

**Discussion Potential:**

4

**Paper Summary:**

The paper argues that sociopolitical risk, or risk to our ability to self-determine through (democratic) institutions, from AI is not always visible from auditing or evaluating the safety of one model.  Instead, we should evaluate how AI alters the conditions of governance.

**Position:**

Yes

**Position In Title:**

Yes

**Related Work:**

3

**Strengths And Weaknesses:**

The paper's central point seems correct -- model-level alignment is insufficient to prevent systemic harms to democratic institutions. The paper contributes to filling a gap between the individual-harms literature and the existential-risk literature, and the framing of "sociopolitical risk" as a distinct category could be a useful conceptual contribution.

Some of the points raised are novel, or at least significantly under-discussed in a machine learning context, such as the potential for use of AI to create bureaucratic congestion (3.3).  Several of the points are illustrated with specific incidents, such as the Biden robocalls, the Zelenskyy deepfake, the GPT-4o sycophancy rollback, Maricopa County FOIA floods.

The "normative centralization" threat model (P6) is a substantive contribution.  The observation that AI infrastructure differs from prior choke points like SWIFT because it carries embedded normative constraints (via model constitutions) is a genuinely interesting point that deserves more attention in the literature.  It relates to Seth Lazar's Governing the Algorithmic City, but makes a different point.

The paper is very well-written and more interesting than many similar works.  It stands out as being a more novel and persuasive argument than some.

Areas of Improvement:

The empirical evidence is illustrative rather than systematic. Most examples are individual incidents or single studies. For a paper claiming that sociopolitical risks are fundamentally about aggregate, population-level effects, it's notable that no population-level evidence is presented. The paper could strengthen its case by pointing to (or calling for) longitudinal or large-scale observational data.

The recommendations themselves are a bit underspecified.  "Institutional Safety Levels" (R1) are proposed but not fleshed out. It could help to have a concrete example of what an ISL threshold looks like quantitatively, how it would be measured, or who would enforce it. Similarly, R4 (pluralistic alignment) recommends running multiple models in parallel but doesn't address the practical costs, the risk of lowest-common-denominator outputs, or how disagreement between models would actually be adjudicated.

Most of the recommendations target policymakers and procurement officials. For the ICML audience, it would help to articulate more clearly what ML researchers should build or measure differently as a result of this framing — specific benchmark designs, evaluation protocols, or technical mechanisms.

The "alternative perspectives" section (Section 5) is brief and not compelling. The Hayekian self-adaptation counterargument is dismissed mainly by asserting that AI moves faster than prior technologies, without much evidence. The alignment-sufficiency counterargument is dismissed by restating the paper's thesis. Engaging more seriously with these objections or different objections would strengthen the paper.

The paper sometimes seems to assume that all currently vulnerable institutions were well-functioning information-processing systems before the introduction of AI. Some might argue that many of the vulnerabilities described (bureaucratic overload, opacity, epistemic flooding) pre-date AI — the paper would be stronger if it engaged with the baseline range in functionality of institutions and argued for why AI represents a difference in kind rather than a difference in degree.

It would be helpful to talk more explicitly about democratic institutions. The word only comes up later in the paper, but surely the sociopolitical risks to hereditary monarchies or autocratic dictatorships (while still present) are different than the ones described. Democracy is indicated by the citations to Landemore, but it would be helpful to mention it earlier. It would also be helpful to cite other democratic theorists in addition to Landemore to support other points.
- Landemore and Scott Page have a collaborative paper about why homogenization degrades decision-making; Hong and Page also support this point, as do other theorists of "epistemic democracy"
- Elinor Ostrom on governing the commons

The paper could also benefit from engaging more with literature on algorithmic monoculture, including the "Ethics of Scale" section from Bommasani et al's "On the Opportunities and Risks of Foundation Models", Kleinberg & Raghavan's Algorithmic Monoculture, and Bommasani et al's Ecosystem Level Analysis of Deployed Machine Learning, which makes a similar point that there are some risks that can only be observed at the ecosystem level but can't be observed from single-model behavior.

The points in P4 could cite existing literature on echo chambers and epistemic bubbles, such as Nguyen's "Escape the Echo Chamber" or a critic of the idea that youtube's selection algorithm is as described, such as Chen, Subscriptions and external links help drive resentful users to alternative and extremist YouTube videos.

Section 3.5 could also cite much more on algorithmic opacity and transparency, including Vredenburgh's The Right to Explanation.

R3 perhaps should be "Increasing Trustworthiness" rather than "Increasing Trust", given the content of R3.

**Support:**

3

---

> ### Author Rebuttal · Authors · 2026-03-31
>
> We thank the reviewer for a generous and constructive review. The specific literature suggestions, the push for more concrete recommendations, and the call to strengthen the alternative perspectives section have all materially improved the paper. We address each point below.
>
> **On population-level evidence.** We agree on the importance of longitudinal or large-scale observational evidence showing, for example, that opinion diversity declines in populations with heavy LLM adoption relative to those without. In the revision, we propose multi-agent simulation as a tractable first step toward such evidence. Agent-based modeling has established precedent in computational social science (Epstein & Axtell, 1996), and recent work shows LLM-powered agents can replicate human behavior in social experiments (Park et al., 2023; Aher et al., 2023). We frame simulation as a complement to the population-level empirical studies the reviewer calls for, and we flag this as an important direction for future work.
>
> **On recommendation specificity.** The revision strengthens ISLs by binding capabilities to mandatory safeguards: the shift from "AI drafts internal memos" to "AI generates sentencing recommendations" should trigger disclosure, reasoning trace retention, mandatory human sign-off, and at higher tiers, external audit. We will add that thresholds should be set through democratic input processes. We acknowledge that specifying quantitative capability thresholds remains an open research problem. On pluralistic alignment: we reframe this as procurement diversification, mandating interoperability standards and multi-provider deployment. The reviewer's questions about costs and disagreement resolution are important open questions we flag as research priorities.
>
> **On what ML researchers should build or measure.** The revision adds three recommendations with direct relevance to the ICML community. R1 proposes multi-agent simulation for institutional stress-testing, with concrete evaluation questions (e.g., how does opinion diversity shift when most simulated participants use the same model?). R2 proposes training methods that go beyond harm avoidance toward pro-social epistemic behavior, arguing that multi-agent training environments where sycophancy leads to worse collective outcomes offer a more direct path than single-agent RLHF optimization. R3 raises the question of when increasing AI autonomy degrades institutional accountability faster than it improves performance, connecting to the tool-AI tradition and recent autonomy taxonomies (Morris et al., 2023). We believe these give the ICML audience concrete technical directions that follow from the paper's framing.
>
> **On alternative perspectives.** The Hayekian counterargument is now presented more fully before being engaged: historical precedent is strong, and decentralized adaptation may outperform centralized design. Our response draws on Acemoglu (2021): the capacity for institutional self-correction may itself be degraded by the dynamics it is expected to resolve, distinguishing the current situation from prior transitions where adaptive machinery remained intact. The alignment-sufficiency counterargument now delineates how alignment can directly help with specific sociopolitical risks (less sycophantic training reduces belief reinforcement; improved chain-of-thought faithfulness strengthens auditability), but some failure modes persist regardless of alignment quality (cost-driven bureaucratic congestion, verification asymmetries at scale), and others are direct consequences of how current alignment methods work (output homogenization, normative capture through model constitutions).
>
> **On baseline institutional dysfunction.** The revision now acknowledges that democratic participation has long favored organized and well-resourced actors (citing Gilens & Page, 2014), and frames our claim around the intensification of these dynamics: AI does not invent these asymmetries but changes the economics of participation and oversight in ways that can deepen them.
>
> **On related literature.** We included the reviewer's literature recommendations and believe that they substantially strengthen the paper.
>
> **New references**
>
> Aher et al. Using Large Language Models to Simulate Multiple Humans and Replicate Human Subject Studies. ICML, 2023.
>
> Epstein and Axtell. Growing Artificial Societies: Social Science from the Bottom Up. Brookings Institution Press and MIT Press, 1996.
>
> Gilens and Page. Testing Theories of American Politics: Elites, Interest Groups, and Average Citizens. Perspectives on Politics, 12(3):564-581, 2014.
>
> Morris et al. Levels of AGI: Operationalizing Progress on the Path to AGI. arXiv:2311.02462, 2023. Published at ICML 2024.
>
> Park et al. Generative Agents: Interactive Simulacra of Human Behavior. UIST '23, 2023.

---

> > ### Author Rebuttal · Reviewer_Cw4V · 2026-04-04
> >
> > I appreciate the revisions proposed and think they would strengthen the paper.  Although my final five points were not responded to, I understand that connecting the paper to additional literature is up to the discretion of the authors. Thanks to the authors for their thoughtful response.

---

### Decision · Program_Chairs · 2026-04-30

**Decision:**

Accept (spotlight)

**Comment:**

All the reviews were uniformly positive and there is substantial benefit in coining the phrase "sociopolitical risks" in an AI context and pointing out that it is a neglected area of safety.

I have little doubt that this work has the potential to both spark debate and provide a handy citation for people looking to work in the field.

Reviews were positive about the quality of writing and the idea itself.

I would encourage the authors to consider the early issues raised by the reviewers and to update the text to avoid future misunderstandings.